# Ultrahigh Spin Filter Efficiency, Giant Magnetoresistance and Large Spin Seebeck Coefficient in Monolayer and Bilayer Co-/Fe-/Cu-Phthalocyanine Molecular Devices

**DOI:** 10.3390/nano11102713

**Published:** 2021-10-14

**Authors:** Jianhua Liu, Kun Luo, Hudong Chang, Bing Sun, Zhenhua Wu

**Affiliations:** 1Institute of Microelectronics of Chinese Academy of Sciences, Beijing 100029, China; liujianhua9@ime.ac.cn (J.L.); luokun@ime.ac.cn (K.L.); changhudong@ime.ac.cn (H.C.); 2College of Microelectronics, University of Chinese Academy of Sciences, Beijing 100029, China

**Keywords:** bilayer metal phthalocyanine, molecular device, spin filter efficiency, giant magnetoresistance, spin Seebeck coefficient

## Abstract

The spin related electrical and thermoelectric properties of monolayer and bilayer MPc (M = Co, Fe, Cu) molecular devices in a parallel spin configuration (PC) and an anti-parallel spin configuration (APC) between the V-shaped zigzag-edged graphene nanoribbon electrodes and the center bilayer MPc molecules are investigated by combining the density functional theory and non-equilibrium Green’s function approaches. The results show that there is an ultrahigh spin filter efficiency exceeding 99.99995% and an ultra-large total conductance of 0.49996G_0_ for FePc-CoPc molecular devices in the PC and a nearly pure charge current at high temperature in the APC and a giant MR ratio exceeding 9.87 × 10^6^% at a zero bias. In addition, there are pure spin currents for CuPc and FePc molecular devices in the PC, and an almost pure spin current for FePc molecular devices in the APC at some temperature. Meanwhile, there is a high SFE of about 99.99585% in the PC and a reserved SFE of about −19.533% in the APC and a maximum MR ratio of about 3.69 × 10^8^% for the FePc molecular device. Our results predict that the monolayer and bilayer MPc (M = Co, Fe, Cu) molecular devices possess large advantages in designing high-performance electrical and spintronic molecular devices.

## 1. Introduction

Molecular spintronic devices have been extensively studied in the past decade, and are a hopeful approach to downsize spintronic devices and are expected to be widely used in high-density data storage and quantum computing [1,2]. A large spin-filter efficiency (SFE) could be generated in molecular spintronic devices [3,4,5]. For example, a significant spin-filtering effect was found in Mn-Pc and Fe-Pc with single-walled carbon nanotube (SWCNT) electrodes [6]; the SFE of the chromium-phthalocyanine molecular device with zigzag graphene nanoribbon (ZGNR) electrodes is nearly 100% in a wide bias voltage region [7]. For a bilayer CuPc molecular device, changing the twist angle between the two molecules could also obtain a high SFE [8]. The spin-dependent hybridization of the electrode and molecular orbitals could cause a large magnetoresistance [9]. A molecular junction made of two MnPc molecules linked by single-walled carbon nanotubes shows perfect spin filter effects and an giant magnetoresistance (GMR) [10]. A cyclooligomeric Mn-phthalocyanine dimer molecular junction shows high-efficiency dual spin-filtering [11]. The interaction of spins with heat currents was studied in MPc (M = Mn, Fe, Co, Ni) molecular devices and the MnPc device exhibits a perfect SFE and thermal-SFE as well as sizeable GMR/thermal GMR effects [12]. The metal phthalocyanine has also been used in photodetectors and nano-porous GaN and CoPc p-n vertical heterojunction bonded as a high-performance self-powered ultraviolet photodetector [13].

In this paper, we studied the spin-related electrical and thermoelectric properties of monolayer and bilayer MPc (M = Fe, Co, Cu) molecular device in the PC and APC with two V-shaped zigzag-edged graphene nanoribbons (ZGNRs) electrodes by employing LDA, LDA + U and the non-equilibrium Green’s function (NEGF) in combination with the density functional theory (DFT) [14,15]. These methods have their advantages over other approaches for nanomaterials with transition metals (TM), Fe in particular. The optimized bond lengths between the 3d metal center and the nearest nitrogen atoms for Fe-/Co-/Cu-phthalocyanines in the gas phase in our NanoDCAL programs [16], LDA and LDA + U, are close to the experimental values [17,18,19]. Comparing these with traditional metal electrode, SWCNTs employed as the electrodes for molecular devices could create covalent bridging between electrodes and conducting molecules and have a better performance [20,21,22,23]. The SWCNTs were chosen as electrodes of a series of 3d transition metal(II) phthalocyanines (MPc, M = Mn, Fe, Co, Ni, Cu and Zn) in Reference [6] and only MnPc and FePc could act as nearly perfect spin filters. Due to their quasi-one-dimensional structure and their excellent electronic transport properties, GNRs have also been a potential candidate for conductive electrodes [23,24,25]. The spin-transport properties of molecular devices constructed using hydrogen–phthalocyanine and transition metal (TM)–phthalocyanine molecules with zigzag graphene nanoribbon (ZGNR) electrodes are investigated in Reference [7] and the results show that there exists a giant magnetoresistance in both the hydrogen–phthalocyanine and TM–phthalocyanine systems. Comparing with the SWCNT electrodes in Reference [6] and the ZGNR electrodes in Reference [7], the V-shaped ZGNR electrodes allow the SFE and MR of our monolayer Co-/Fe-/Cu-phthalocyanine molecular devices under equilibrium states to exceed 99.998%/99.9958%/97% in the PC and 2.59 × 10^7^%/2.58 × 10^7^%/3.069 × 10^8^%, respectively. The SFE of the bilayer CoPc and FePc devices exceed 99.9998% and the SFE of the bilayer CuPc molecular device is over 99.99996%, which exceeds those of the bilayer FePc, bilayer CoPc and FePc-CoPc molecular devices. In addition, the SFE of the CuPc-FePc and CuPc-CoPc devices exceed 99.999997%, which are perfect SFEs. In addition, the spin-down and total conductance and SFE of our bilayer Co-/Fe-/Cu-phthalocyanine molecular devices in the PC are bigger than the ones in our monolayer Co-/Fe-/Cu-phthalocyanine molecular devices, the spin-down channel dominates the transmission and density of state at Fermi energy in the PC. There are large pure spin currents in the monolayer CuPc and FePc molecular devices in the PC at some temperatures. Additionally, the spin-dependent charge Seebeck coefficient and the spin Seebeck coefficient are almost equal for CoPc, CuPc-CuPc, CoPc-CoPc, FePc-FePc, CuPc-CoPc, CuPc-FePc, and CoPc-FePc molecular devices in the PC due to the spin-down Seebeck coefficient being close to zero and there are usually large charge Seebeck coefficients at high temperatures for the abovementioned molecular devices in the APC. Physical mechanisms are proposed for these phenomena. The calculated transmission spectra and the real-space scattering states of the modeled mono- and bilayers can be used for future experimental photoemission spectroscopy and other studies. These theoretical calculations predict new monolayer and bilayer phthalocyanine-based molecular devices with high MR and Seebeck coefficients. Based on the calculated spin-dependent conductance and other characteristics, ultrahigh SFE, GMR, and Seebeck coefficients for Co-/Fe-/Cu-phthalocyanine molecular devices are predicted for the first time. This makes the results motivating for further experimental studies of phthalocyanine- and TM-layered structures. The obtained large SFE of the parallel spin configuration can be utilized in various molecular spintronic devices.

## 2. Methods

We investigated the spin transport properties of monolayer and bilayer MPc (M = Fe, Co, Cu) molecular devices by combining the density functional theory and non-equilibrium Green’s function approach, as implemented by the NanoDCAL transport package [16,26]. Figure 1a and b show top and side views of the structure of themonolayer and bilayer MPc (M = Cu, Fe, Co) molecular devices. The device was formed of three parts: left and right electrodes (which extended to ±∞) and the central scattering region, which contained MPc molecules, in the left and right buffer layers. The energy cutoff was set to 150 Rydberg, the k-point grid was set to 100 × 1 × 1 and electrode temperature was chosen to be 300 K. The electrodes of ZGNRS were modeled with a supercell (4.9332 Å × 32.821 Å × 18 Å for the monolayer MPc and 4.9332 Å × 32.821 Å × 18 Å for the bilayer MPc, the distance between the bilayer MPc was 3 Å) subjected to periodic boundary conditions. A vacuum layer about 15 Å in the y direction and z direction was introduced to eliminate interactions between GNRs in neighboring cells and the edge atoms; both electrodes and central region were saturated with hydrogen (H) atoms to remove the dangling bonds [23]. The exchange-correlation function is described by the local density approximation (LDA) proposed by Perdew and Zunger, and a plus U correction [27] (LDA + U) was used for the above calculations, considering the localized 3d-orbital of Fe (U = 3 eV) and Co (U = 2 eV) atoms [28]. Quantum transport phenomena with the monolayer and bilayer MPc molecular devices were further understood by analyzing the transmission spectrum, projected density of states, and the scattering states.

Spin-polarized zero-bias conductance is given by the Landauer–Buttiker formula [29]
(1)$$G_{\sigma }=\frac{e^{2}}{h}T_{\sigma }\left(E_{F}\right)\;$$
where $T_{\sigma }\left(E_{F}\right)$ is the electron transmission coefficient for the spin-up (↑) and spin-down (↓) electrons (σ=↑/↓) and $E_{F}$ is the Fermi level.

At zero bias, SFE is defined as:(2)$$SFE=\frac{T_{↓}\left(E_{F}\right)-T_{↑}\left(E_{F}\right)}{T_{↑}\left(E_{F}\right)+T_{↓}\left(E_{F}\right)}\times 100\%\;$$
where $T_{↑}\left(E_{F}\right)$ and $T_{↓}\left(E_{F}\right)$ stand for the transmission coefficient of the spin-up (SU) and spin-down (SD) states at the Fermi level, respectively.

Considering the spin direction of the lead, there is a giant magnetoresistance (MR) effect and the MR ratio for the device for the PC and APC are defined as:(3)$$MR=\frac{T_{PC}-T_{APC}}{T_{APC}}\times 100\%\;$$
where $T_{PC}=T_{PC}{\left(E_{F}\right)}_{↑}+T_{PC}{\left(E_{F}\right)}_{↓}$*,*
$T_{APC}=T_{APC}{\left(E_{F}\right)}_{↑}+T_{APC}{\left(E_{F}\right)}_{↓}$.

Considering the spin-dependent thermoelectric properties of the monolayer and bilayer MPc (M = Fe, Co, Cu) molecular devices, the usual charge Seebeck coefficient (S_C_) and the spin Seebeck coefficient (S_S_) are defined as S_C_ = (S_↑_ + S_↓_)/2 and S_S_ = (S_↑_ − S_↓_)/2, where S_↑_ and S_↓_ are the SU and SD Seebeck coefficients, respectively [30].

## 3. Results and Discussion

By comparing the optimized bond lengths between the 3d metal center (M) and the nearest nitrogen (N) atoms for Fe-/Co-/Cu-phthalocyanines in the gas phase in our NanoDCAL program LDA (for CuPc) and LDA + U (for FePc and CoPc) results with calculations for the same systems done with the SIESTA + SMEAOGOL programs and the PBE and PBEh functionals in Reference [6] as well as the experimental values in Table 1. We found that our results were close to the experimental values and our NanoDCAL program LDA(+U) was very characteristic for Fe-/Co-/Cu-phthalocyanines molecular devices.

For monolayer CoPc, FePc, CuPc, bilayer CuPc-CuPc, FePc-FePc, CoPc-CoPc, FePc-CoPc, CoPc-CuPc and CuPc-FePc molecular devices (MDs) at zero bias, Figure 2a shows that the spin-up (SU) conductance of the MPc for the PC is notably smaller than the spin-down (SD) one. The SD G and the total G of the FePc-CoPc MD and SU G of CuPc MD in the PC, total G of CuPc-FePc MD, SU G of CuPc-CoPc MD and SD G of CuPc-CuPc MD in the APC are maximum for the above MPc(s) MDs, respectively. The SD G and total G of bilayer MPc(s) MD are larger than the ones of the monolayer MPc MD in the PC and APC, respectively. Based on Equation (1), the change rule of the transmission at the Fermi level of the MPc(s) MD is perfectly consistent with that of the conductance of the MPc(s) shown in Figure 2b. The physical mechanism of the conductance change law can be understood by analyzing the transmission spectra, scattering states, and projected density of state as follows:

For CoPc, CuPc, FePc, CuPc-CuPc, FePc-FePc, CoPc-CoPc, FePc-CoPc, CuPc-CoPc, and FePc-CuPc molecular devices at zero bias, the SFE in the PC are all close to 100% and there is a maximum SFE of 99.999998% for CuPc-CoPc MD in Figure 3. In the APC, the SFE of all MPc MDs are relatively small and there are negative SFEs for CuPc, FePc, FePc-FePc, CoPc-CoPc, CuPc-CoPc and FePc-CuPc MDs, which means that their SU transmission is greater than SD transmission and the spin polarization is reversed. There is a minimum SFE of about −24.183% for FePc-FePc MD. The SFE is determined by the transmission in the SU and SD channel at the Fermi level based on Equation (2). Figure 3 also shows that there is a giant MR effect in MPc(s) MD. There is a maximum MR ratio of about 3.69 × 10^8^% for the monolayer FePc MD, and the MR ratio of FePc-FePc MD is largest in the bilayer MPcs MD.

To better understand the transport properties of the MPc molecular devices, we investigated transmission and the projected density of state (projected onto orbitals with considering the angular momentum quantum number, that is AMQN) in the SU and SD channels among FePc and FePc-FePc molecular devices in the PC and APC under equilibrium states, as shown in Figure 4 (there are two main SU and three main transmission peaks from −1 eV to 1 eV at zero bias: −0.82 eV, 0.08 eV, 0.99 eV for SD and the SU transmission peak is −0.82 eV, 0.98 eV). Owing that the origin of the transmission peaks could be figured out by projected density of states (PDOS) [31], as shown in Figure 4a, there were several obvious PDOS peaks of p- and d-orbitals, some of them corresponded basically to the transmission peaks at or around the same energy. For the FePc MDs in the APC shown in Figure 4c, the corresponding situation of the PDOS peaks and transmission peaks is similar to the one in Figure 4a. However, Figure 4b shows that there are many PDOS peaks of s-, p- and d-orbitals, and there are only three obvious SU transmission peaks and two obvious SD transmission peaks for FePc-FePc MDs in the PC, therefore, just the PDOS peaks at or around the transmission peak make a contribution to the transmission, and the FePc-FePc MD in the APC appears the same situation and is shown in Figure 4d. The transmission is mainly controlled by the PDOS of the p orbital.

Figure 5 shows the real-space scattering states of FePc and FePc-FePc molecular devices in the PC and APC at zero bias. Figure 5a demonstrates that the SD scattering of incoming state of lead L (SSLL) and lead R (SSLR) of the FePc MD in the PC are larger than the SD one, which corresponds to the fact that the SD G is greater than the SU G of the FePc MD in the PC. The SU SSLL and SSLR of the FePc MD in the PC shown in Figure 5a is smaller than the ones in the APC shown in Figure 5b, the SD SSLL and SSLR of the FePc MD in the PC is larger than the ones in the APC, which corresponds the SU G of the FePc MD in the PC being lower than the one in the APC and the SD G of the FePc MD in the PC is greater than the one in the APC. Figure 5c,d shows a similar situation for FePc-FePc MDs in the PC and APC. Meanwhile, the SD SSLL and SSLR of FePc-FePc MDs in the PC and APC are greater than the ones of FePc in the PC and APC, which also explains why the G of the FePc-FePc MD is larger than the one of the FePc MD in the PC and APC.

Figure 6a,b demonstrates the spin-dependent Seebeck coefficient S_σ_ in the left column, the spin-dependent charge Seebeck coefficient S_C_ and the spin Seebeck coefficient S_S_ in the right column of CuPc, CoPc, FePc MD in the PC and APC versus the temperature at zero bias. Generally, shown in Figure 6a, for the S_↓_ of CuPc and CoPc in the PC, the latter is quite small, S_↓_ ≈ 0. The S_↑_ of CuPc, CoPc, FePc MD in the PC first decreases, then increases and then decreases with T. S_C_ = 0 at T ≈ 307 and 482 K for CuPc MD and S_C_ = 0 at T ≈ 218 K for FePc MD in the PC, a pure spin current produced by the temperature gradient. For CoPc MD in the PC, S_C_ ≈ S_S_ duo to S_↓_ is close to zero and described by the formula of S_C_ and S_S_. As shown in Figure 6a, at a low temperature, T, S_σ_ is linear in T [32] and there are large and negative S_↑_ and S_↓_ around T = 160 K for CuPc MD, large S_↑_ and S_↓_ around T = 211 and 261 K for CoPc MD, and a large S_↓_ around T = 207 K and a large and negative S_↑_ around 207 K for FePc MD in the APC. There is a pure spin current with S_C_ = 0 at T ≈ 100 K for CuPc MD, at T ≈ 143 K for CoPc MD, at T≈99 K for FePc MD and a large and negative S_C_ and a small and negative S_S_ at T = 161 K for CuPc MD, a large S_C_ and a small S_S_ at T = 500 K for CoPc MD, a large and negative S_S_ and a small and negative S_C_ at T = 206 K for FePc MD in the APC.

Figure 7a,b shows that the S_σ_ in the left column, the S_C_ and S_S_ in the right column of CuPc-CuPc, CoPc-CoPc, FePc-FePc, CuPc-CoPc, CuPc-FePc, CoPc-FePc MD in the PC and APC versus the temperature at zero bias. The S_↓_ of the bilayer MPc MD is close to zero and S_C_≈S_S_ in the PC in Figure 7a,b. Interestingly, the S_S_ is close to zero in most regions of the temperature range. There is a tendency to produce a pure charge current with the increasing temperature and S_↓_≈S_↑_ for CoPc-CoPc, CuPc-CoPc, CuPc-FePc, CoPc-FePc MDs in the APC, as shown in Figure 7c,d.

Therefore, we can obtain an ultrahigh spin filter efficiency exceeding 99.99995% and ultra-large total conductance of 0.49996G0 for FePc-CoPc molecular device in the PC and a nearly pure charge current at the high temperature in the APC as well as a giant MR ratio exceeding 9.87 × 10^6^% at zero bias. There are pure spin currents for CuPc and FePc molecular devices in the PC, and an almost pure spin current for FePc molecular device in the APC at some temperatures. Meanwhile, there is a high SFE of about 99.99585% in the PC and a reserved SFE of about −19.533% in the APC as well as a maximum MR ratio of about 3.69 × 10^8^% for monolayer FePc molecular device. In addition, the spin-dependent charge Seebeck coefficient S_C_ and the spin Seebeck coefficient S_S_ are almost equal for CoPc, CuPc-CuPc, CoPc-CoPc, FePc-FePc, CuPc-CoPc, CuPc-FePc, CoPc-FePc molecular devices in the PC duo to the spin-down Seebeck coefficient is close to zero and there are usually large charge Seebeck coefficients at high temperatures for the above molecular devices in the APC.

## 4. Conclusions

In conclusion, we investigated the spin-dependent conductance, spin filter efficiency, giant magnetoresistance ratio, Seebeck coefficient, charge Seebeck coefficient and spin Seebeck coefficient by analyzing the projected density of state, transmission spectrum and scattering state of monolayer and bilayer MPc (M = Fe, Co, Cu) molecular devices in the parallel and anti-parallel spin configurations by employing LDA, LDA + U and nonequilibrium Green’s function approaches. These methods have their benefits over other approaches for nanomaterials with transition metals, Fe in particular. The results show that the spin filter efficiency in the parallel spin configuration, spin-down and total conductance of the bilayer MPc molecular devices were superior to the monolayer MPc molecular device. There are large pure spin currents in the CuPc and FePc molecular devices in the parallel spin configuration at some temperatures. Meanwhile, there is a high SFE of about 99.99585% in the parallel spin configuration and a reserved SFE of about −19.533% in the anti-parallel spin configuration and a maximum MR ratio of about 3.69 × 10^8^% for monolayer FePc molecular devices. These transport phenomena could be well understood by analyzing the transmission spectra, projected density of state and scattering states. The calculated transmission spectra and the real-space scattering states of the modeled mono- and bilayers can be used for future experimental photoemission spectroscopy and other studies. These theoretical calculations predict new monolayer and bilayer phthalocyanine-based molecular devices with high MR and Seebeck coefficients. This makes the results motivating for further experimental studies of phthalocyanine and transition metal layered structures. The obtained large spin-filter efficiency of the parallel spin configuration can be utilized in various molecular spintronic devices.

## Figures and Tables

**Figure 1 nanomaterials-11-02713-f001:**
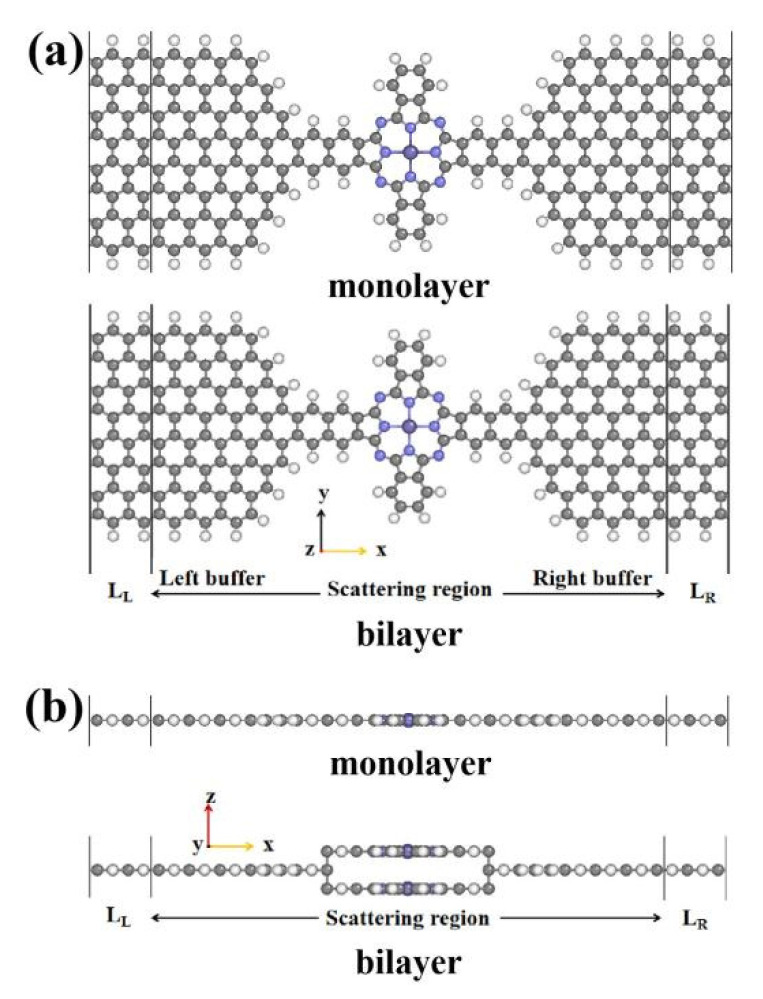
The structure of the monolayer and bilayer MPc (M = Cu, Fe, Co) molecular devices. The navy, blue, black, and grey balls represent M, N, C, and H atoms, respectively. The molecule in the scattering region is MPc, the electrode is V-shaped zigzag-edged GNRs. (**a**,**b**) are the top and side view of the molecular device.

**Figure 2 nanomaterials-11-02713-f002:**
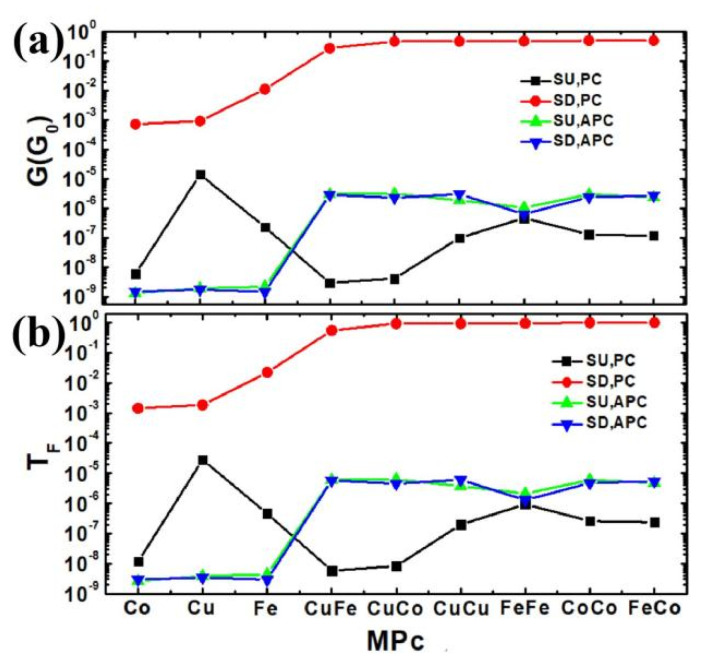
The conductance G (**a**) and transmission at the Fermi level (**b**) of the MPc molecular device for monolayer CoPc, CuPc, FePc, bilayer CuPc-FePc, CuPc-CoPc, CuPc-CuPc, FePc-FePc, CoPc-CoPc, and FePc-CoPc at zero bias.

**Figure 3 nanomaterials-11-02713-f003:**
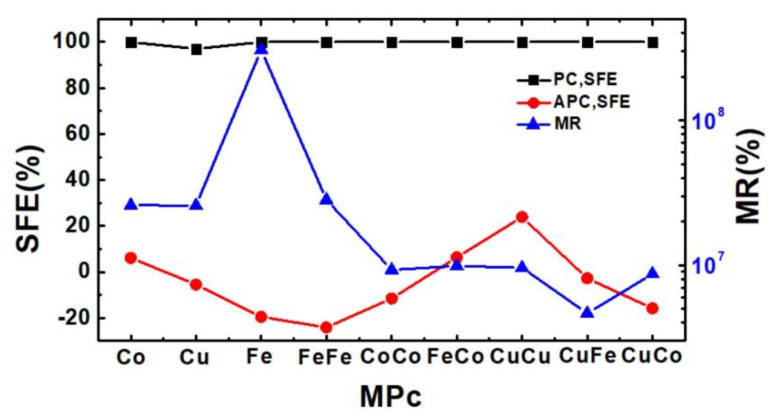
The SFE and MR of the MPc molecular device for the monolayer CoPc, CuPc, FePc, bilayer CuPc-FePc, CuPc-CoPc, CuPc-CuPc, FePc-FePc, CoPc-CoPc, and FePc-CoPc at zero bias.

**Figure 4 nanomaterials-11-02713-f004:**
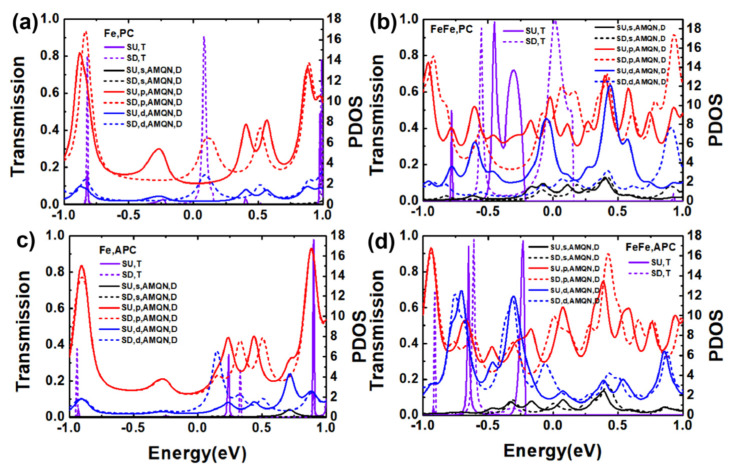
The spin-dependent transmission (T) and projected density of state (D) for s-, p-, d-orbital of the scattering region of the FePc and FePc-FePc MD in the PC (**a**,**b**) and APC (**c**,**d**) at zero bias.

**Figure 5 nanomaterials-11-02713-f005:**
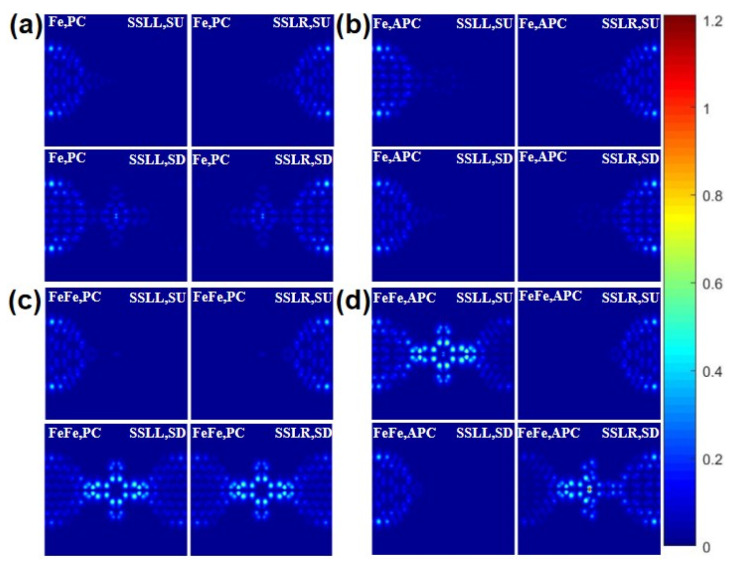
The scattering states of the FePc in the PC (**a**) and APC (**b**), and FePc-FePc MDs in the PC (**c**) and APC (**d**) at zero bias.

**Figure 6 nanomaterials-11-02713-f006:**
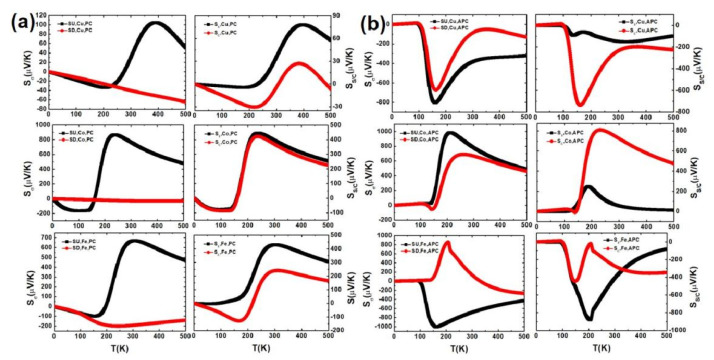
The spin-dependent Seebeck coefficient S_σ_, charge Seebeck coefficient S_C_ and the spin Seebeck coefficient S_S_ of CuPc, CoPc, FePc MD in the PC (**a**) and APC (**b**) at zero bias.

**Figure 7 nanomaterials-11-02713-f007:**
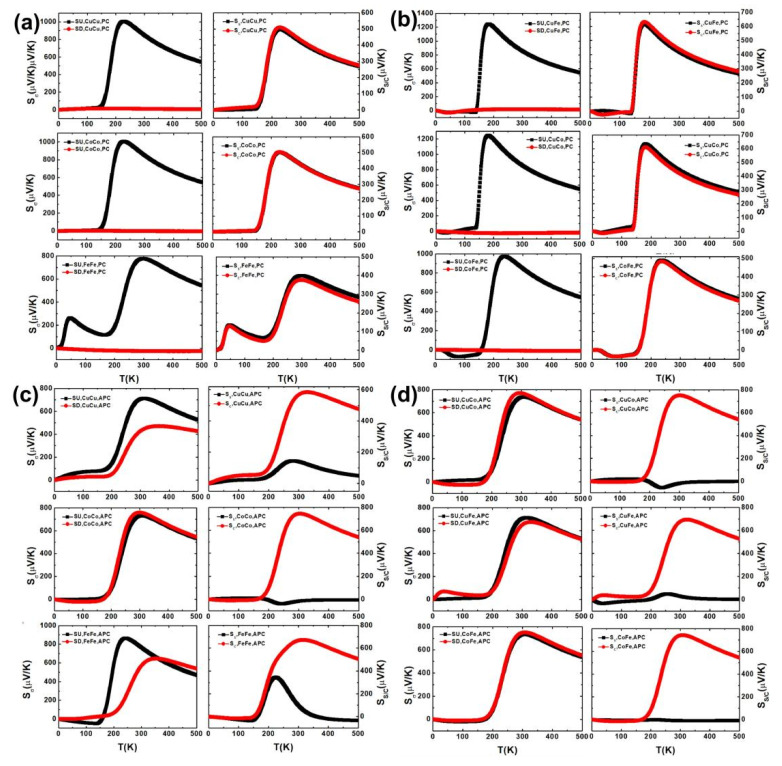
The spin-dependent Seebeck coefficient S_σ_, charge Seebeck coefficient S_C_ and the spin Seebeck coefficient S_S_ of CuPc-CuPc, CoPc-CoPc, FePc-FePc, CuPc-CoPc, CuPc-FePc, CoPc-FePc MDs in the PC (**a**,**b**) and APC (**c**,**d**) at zero bias.

**Table 1 nanomaterials-11-02713-t001:** Optimized bond lengths between the 3d metal center (M) and the nearest nitrogen (N) atoms for Fe-/-Co-/-Cu-phthalocyanines in the gas phase. The experimental values are also given for comparison.

	M–N (Å)				
		Gaussian 03			
MPc molecule	SIESTA Ref. [6]	PBE Ref. [6]	PBEh Ref. [6]	LDA(+U) Our work	Exp.
FePc	1.943	1.934	1.941	1.937(+U)	1.927 (Ref. [17])
CoPc	1.931	1.934	1.932	1.893(+U)	1.908 (Ref. [18])
CuPc	1.978	1.967	1.954	1.936	1.932 (Ref. [19])

## Data Availability

The data is available on reasonable request from the authors. The data are not publicly available due to ethical.

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
