# Peer review of "Ultrahigh Spin Filter Efficiency, Giant Magnetoresistance and Large Spin Seebeck Coefficient in Monolayer and Bilayer Co-/Fe-/Cu-Phthalocyanine Molecular Devices"

_nanomaterials, 2021, doi:10.3390/nano11102713_

Round 1

Reviewer 1 Report

The study of conductivity through individual molecules, including their spintronic properties, holds promise for future electronics applications and has been studied for over a decade now.  This manuscript is a contribution to the field of molecular spintronics.  Although there are problem with the English and some of these lead to phrases which I find completely incomprehensible, the manuscript is by and large written in a pleasant and polished way.  However I see major problems which prevent me from recommending acceptance at this time.  The major difficulty is that the authors (i) fail to make clear what makes their work distinct from and an advance over previous work in the field and (ii) fail to give an adequate comparison of their work with previous work in the field.  Another major difficulty is that the authors
do not give an adequate justification of their choice of methodology and have not convinced me that they have run all of the checks that they should have done before reporting their calculations.  I will mention Ref. [7] as one example of an apparently similar study with which comparisons should have been made:

[7] Shen, X.; Sun, L.L.; Yi, Z.L.; Benassi, E.; Zhang, R.X.; Shen, Z.Y.; Sanvito, S.;
Hou, S.M. "Spin transport properties of 3d transition metal (II) phthalocyanines in contact with single-walled carbon nanotube electrodes." Phys. Chem. Chem. Phys. 2010, 12, 10805. DOI: 10.1039/C002301A

Ref. [7] may also be taken as a good model of how a scientific paper should be written.

One of the jobs of a referee is "simply" to check if he or she believes the reported work and if there is enough information given to be able to reproduce the calculations. One of the issues in these types of calculations is whether the results will change when parts of the electrodes are included in the central region of the model.  If they do, then the results are not reliable until enough of the electrode is included in the central region that the results have stopped changing.  The authors need to do this or, if they have done so, they need to discuss the results of this validation step.  Another technical issue is the choice of density functional.  The authors used the LDA+U approach which uses
a semiempirical parameter U whose value needs to be justified, or at the very least motivated. The authors fail to do this.  As a general rule the choice of density functional needs to be justified by testing it, for related molecules and properties, against experiment and/or high-quality calculations.  At the very least, the authors need to compare and discuss their NanoDCAL (not Nanodcal!) program LDA+U results with calculations for the same systems (!) done with the SIESTA + SMEAOGOL programs and the PBE and PBEh functionals in Ref. [7] Are the results similar or very different?  If they are similar, then that is reassuring.  If they are very different, then which is more credible and why? Another issue is whether the NanoDCAL minimal Fireball basis set is still state-of-the-art for this type of calculation.  I believe that the authors' results are technically correct but I am not convinced that they have made a serious evaluation of the reliability of their choice of model.

In fact, their manuscript reads like they obtained a program, carried out all the
calculations they could, and reported eveything that they found.  That is not how you should write a scientific paper!  At the very least, this manuscript requires a major rewrite to bring out what makes their work different and new compared to previous work (such as Ref. [7]).  They should separate their manuscript into a validation part which compares those results which are not novel with existent experimental and computational results in the literature and a new results part which emphasizes what this work contributes to the advancement of knowledge in this field and hence why it deserves to be published.  Naturally this should be accompanied by a proper explanation in the introduction explaining why this new contribution is going to be interesting and some emphasis in the conclusion of what is new that has been learned and why this could be important.  All of this is missing.

Until the above problems are remedied, I cannot see myself recommending acceptance ofthis manuscript.

QUESTIONS AND COMMENTS:

1) Parts (a) and (b) seem to be identical.  Hence G(G_0) = T_F . Only one graph, along with the explanation in the text, is needed to communicate the necessary information.

2) Most nanodevices of this type are actually short lived as they are found
experimentally to rapidly burn out due to Joule heating.  The authors should
comment on this.

3) Page 4, what is E11?  I think that this needs to be defined more carefully.

6) The NanoDCAL web page should be cited: https://www.nanoacademic.com/product-page/nanodcal

ENGLISH:

1) Title, element names are not usually capitalized in English
2) Abstract, line 3, "graphene nanoribbon electrodes"
3) Abstract, "by combining the density-functional theory and
nonequilibrium Green's function approaches"
4) Abstract, what is a "well-performance charge current"?  This English
is incomprehensible.
5) Page 1, 6 lines from the bottom, "A cyclooligomeric Mn-phtalocyanine dimer"
6) Page 2, line 49, "density-functional theory (DFT)" (missing space)
7) Page 2, line 52, "channel dominates the transmission"
8) Page 2, line 71, "4.9332 Å x 32.821 Å x 18 Å"
9) Page 2, line 79, "U = 3 eV" and "U = 2 eV"
10) Page 2, remove the indent just after Eq. (1) and remove the indent just after Eq. (2).  (This type of error is typical of first time users of LaTeX.  It comes from putting a blank line after the equation environment which should not be there and seems to be a common problem in this manuscript.)
11) Page 4, lines 137 and 138, there should be a space between the number and the units, so "0.08 eV" not "0.08eV"
12) Page 6, lines 178-182, separate units from numbers so "207 K" rather than "207K"
13) Page 6, line 188, "is close to zero in most of the temperature range. There is"
14) Page 7, line 198, what is a "well-performance charge current"?  (A search of the internet indicates that "well performance" means how well a well preforms, which makes no sense here.)
15) "Nanodcal" should be typed as "NanoDCAL"

Reviewer 2 Report

In this manuscript by Jianhua Liu et al. from Institute of Microelectronics of Chinese Academy of Sciences and University of Chinese Academy of Sciences, Beijing, China, the monolayer and bilayer MPc (M=Fe, Co, Cu) molecular devices were investigated employing LDA, LDA+U and nonequilibrium Green's functions approaches.

These methods have large advantages over other approaches for nanomaterials with transition metals (TM), Fe, in particular. The calculated transmission spectra and the real-space scattering states of the modeled mono- and bilayers can be used for future experimental photoemission spectroscopy and other studies. These theoretical calculations predict new monolayer and bilayer phthalocyanine-based molecular devices with high MR and Seebeck coefficient, for this reason, the work is relevant to nanomaterials. Based on the calculated spin-dependent conductance and other characterictics, ultrahigh SFE, giant magnetoresistance ratio and Seebeck coefficients for the Co / Fe / Cu - phthalocyanine molecular devices are predicted for the first time. This makes the results motivating for further experimental studies of phthalocyanine and TM layered structures. The obtained large spin-filter efficiency (SFE) of the parallel spin configuration (PC) can be utilized in various molecular spintronic devices. These issues can be added to the text: 

  • LDA+U method is used for Fe (U=3eV) and Co (U=2eV). Why these values were taken? Additional references are required. 
  • PBE (GGA) exchange-correlation approximation is very common. Could it be used in these calculations?
  • Spin-orbit coupling (SOC) can affect the magnetic characteristics of Co compounds. Please comment on this. 

The manuscript can be published in Nanomaterials after minor revision. 

Round 2

Reviewer 1 Report

I refereed the initial submission of this article which I found to be interesting and also to be written in an acceptable English.  However it was badly written because (1) it did not adequately motivate the study and explain why it was different from previous studies and (2) I did not find it convincing that the authors had sufficiently validated their methodology.  I asked for a major rewrite.  I would say that the authors have met my request half way and I will grudgingly accept what they have done.

I say "grudgingly" because I found both their coverletter and their major changes to be evasive.  Either they do not fully understand the issues that I raised or they are deliberately chosing to ignore them.  I do not know which.  I am glad that they added a comparison of geometries calculated with different functionals and programs, but I would have been even happier had they included a comparison of band energies (band gap!) with experiment.

In my comments on the initial submission, I noted that, "The authors used the LDA+U approach which uses a semiempirical parameter U whose value needs to be justified, or at the very least motivated.  The authors fail to do this." The only response that the authors seem to have made to this is to add a reference in their sentence: "The exchange-correlation function is described by local density approximation (LDA) proposed by Perdew and Zunger, and a plus U correction [18,27] (LDA+U) is used for the above calculations considering the localized 3d-orbital of Fe (U=3 eV) and Co (U=2 eV) atoms [28]."The usual way to fix U is to adjust it until the band structure improves. I looked carefully through Ref. [28]:

Tao, L.L.; Wang, J. Ferroelectricity and tunneling electroresistance effect
driven by asymmetric polar interfaces in all-oxide ferroelectric tunnel
junctions. Appl. Phys. Lett. 2016, 108, 062903. DOI: 10.1063/1.4941805

There is no indication of how U was chosen.  In fact, Ref. [28] is so bereft of technical details that it would be difficult, if not impossible, to reproduce their calculations.  What a bad example of scientific writing!

In the end, I think I will leave it up to the editor to decide if this is up to the scientific standards of Nanomaterials because I still have some lingering doubts.

Author Response

Please see the attachment."
